# Temporal Changes in Association Patterns of Cattle Grazing at Two Stocking Densities in a Central Arizona Rangeland

**DOI:** 10.3390/ani11092635

**Published:** 2021-09-08

**Authors:** Colin T. Tobin, Derek W. Bailey, Mitchell B. Stephenson, Mark G. Trotter

**Affiliations:** 1Department of Animal and Range Sciences, New Mexico State University, Las Cruces, NM 88003, USA; dwbailey@nmsu.edu; 2Panhandle Research and Extension Center, University of Nebraska—Lincoln, Scottsbluff, NE 69361, USA; mstephenson@unl.edu; 3School of Medical and Applied Sciences, Central Queensland University, Rockhampton, QLD 4702, Australia; m.trotter@cqu.edu.au

**Keywords:** GPS tracking, grazing distribution, forage utilization, social associations, cattle

## Abstract

**Simple Summary:**

Monitoring changes in the utilization of forages across rangelands can be time consuming and difficult with untrained personnel. The use of real time positioning for cattle is becoming commercially available with the improvements in technology. The objective of this case study was to identify the changes in livestock social associations and spatial location at two stocking densities throughout a six-week grazing period. Both pastures used similar sized herds with 35 and 29 animals tracked with global positioning systems set at 30-min intervals. A half-weight index value was calculated for each pair of tracked cattle to determine the proportion of time that cattle were within 75 m and 500 m of each other. Throughout the study, forage utilization increased from 5 to 24% and from 10% to 20% and forage mass decreased from 2601 kg ha^−1^ to 1828 kg ha^−1^ and 2343 kg ha^−1^ to 1904 kg ha^−1^, in the high stocking density pasture and low stocking density pasture, respectively. Utilization of forages throughout the trial forced cattle to disperse and travel further from water sources to find new feeds. Real-time GPS tracking has the potential to remotely detect changes in animal spatial association, identify when cows disperse, and improve recognition for the need of pasture rotation to avoid rangeland degradation.

**Abstract:**

Proper grazing management of arid and semi-arid rangelands requires experienced personnel and monitoring. Applications of GPS tracking and sensor technologies could help ranchers identify livestock well-being and grazing management issues so that they can promptly respond. The objective of this case study was to evaluate temporal changes in cattle association patterns using global positioning system (GPS) tracking in pastures with different stocking densities (low stocking density [LSD] = 0.123 animals ha^−1^, high stocking density [HSD] = 0.417 animals ha^−1^) at a ranch near Prescott, Arizona. Both pastures contained similar herd sizes (135 and 130 cows, respectively). A total of 32 cows in the HSD herd and 29 cows in the LSD herd were tracked using GPS collars at location fixes of 30 min during a 6-week trial in the summer of 2019. A half-weight index (HWI) value was calculated for each pair of GPS-tracked cattle (i.e., dyads) to determine the proportion of time that cattle were within 75 m and 500 m of each other. Forage mass of both pastures were relatively similar at the beginning of the study and forage utilization increased from 5 to 24% in the HSD pasture and increased from 10 to 20% in the LSD pasture. Cattle in both pastures exhibited relatively low mean association values (HWI < 0.25) at both spatial scales. Near the end of the study, cattle began to disperse likely in search of forages (*p* < 0.01) and travelled farther (*p* < 0.01) from water than during earlier periods. Real-time GPS tracking has the potential to remotely detect changes in animal spatial association (e.g., HWI), and identify when cows disperse, likely searching for forage.

## 1. Introduction

The availability of global positioning systems (GPS) since the late 1980s has greatly improved our understanding of wild and domesticated animal behavior [1]. Improvement of GPS technology and analysis software has facilitated experiments conducted on rangelands with free roaming livestock as opposed to studies using pens or small pastures in controlled settings [2]. Management of livestock grazing on extensive rangeland landscapes has different challenges than livestock managed in intensive systems [3]. Rangelands tend to be rugged and expansive [4], making observation of individual animals difficult compared to intensive systems (e.g., feedlot or dairy) where livestock can be observed several times per day. The ability to understand changes in animal behavior due to management and human influence in rangeland livestock production conditions can improve animal welfare, productivity, grazing distribution, and watershed health [3,4,5,6].

Livestock grazing distribution is one of the four principles of grazing management [7] and manipulation of livestock spatial movement patterns is critical for the environmental and economic sustainability of rangeland livestock operations [8]. Forage quality and quantity, terrain, and social interactions can affect distribution of livestock across rangelands [9,10]. However, the impact and consequences of social interactions on livestock grazing patterns is understudied and enhancing our knowledge would improve our ability to manage, thus deserving further study [11,12].

Social hierarchies are evident in small, well established groups of livestock, where dominance can occur [13,14,15,16,17]. Šárová et al. [18] found that dominant animals tended to lead herds during travelling and foraging bouts. This social dominance may be weakened during frequent mixing and rotation of pastures not allowing for isolation of animals [19]. Examining social hierarchies and relationships among herd members may provide insight into distribution patterns on extensive rangelands.

Previous studies involving cattle have addressed questions regarding social interactions [17], social dominance [18,20], relationships among cows in variable herd sizes [16], and relationships among cows during different seasons of use [21]. However, few studies have addressed the impact of forage availability and utilization that can occur due to differing stocking densities on the associations among cows within similar sized herds. Livestock associations may shift as forage availability and quality change as the result of grazing. With higher levels of forage quantity, cattle tend to graze closer together and function differently than when forage is scarce and cattle are required to search for more forage [20,22,23]. Increased forage utilization and the corresponding reductions in available forage biomass in preferred areas is a driver of livestock distribution. For example, as cattle utilize vegetation closer to water at higher rates, they typically begin to use areas farther from water where availability of forage is greater because it has yet to be grazed [24]. Social interactions can be used as indicators of livestock well-being [12] and monitoring cattle social interactions in real time may be useful for understanding how cattle respond to forage depletion in arid and semi-arid rangelands.

Studies were conducted with herd sizes that were large enough (130 and 135 cows per herd) to display social interactions typical of commercial rangeland livestock operations. Stephenson et al. [16] found that spatial associations differed between small (<40 cows) and larger herds (>40 cows). Changes in animal dispersion could potentially indicate changes in animal welfare and social interactions that occur as forage availability decreases as the result of grazing. With the development of real-time GPS systems [4], producers could also potentially monitor changes in social interactions and associations to help monitor forage conditions and determine when cattle should be moved to a new pasture.

The objective of this case study was to evaluate changes in the associations among individual cows during the grazing season at two stocking densities where forage utilization levels increased at different rates over time.

## 2. Materials and Methods

### 2.1. Study Site and Environment

This study was conducted at the Deep Well Ranch (DWR). The DWR is located 16 km north of Prescott, Arizona, United States (112°29′ W, 34°41′ N) and encompasses 8004 ha of rolling terrain that vary in elevation from 1434 to 1657 m. The Köppen climate classification for Prescott, Arizona is Hot-summer Mediterranean (Csa). The average annual precipitation is 450 mm with over 40% occurring during the summer monsoon season (July through September). Vegetation at DWR is predominantly perennial grasslands dominated by black grama (*Bouteloua eripoda* (Torr.) Torr.), dropseeds (*Sporobolous* spp.) and purple threeawn (*Aristida purpurea* Nutt).

Two pastures were utilized for cattle grazing during the study. The low stock-density pasture (North Pasture) contained 1096 ha and varied in elevation from 1471 to 1542 m. The smaller high stock-density pasture (North Ditch Pasture) contained 312 ha and varied in the elevation from 1460 to 1520. The study areas are adjacent to one another and are separated by Arizona State Highway 89.

### 2.2. Animals

A herd of 135 Corriente cow-calf pairs grazed the low stock-density pasture from late May 2019 to mid-August 2019. A separate herd of 130 Corriente cow-calf pairs grazed the high stock-density pasture from late May 2019 until mid-July 2019. As part of their normal management, the ranch has two separate herds that graze on the west and east side of Arizona Highway 89. Cows varied in age from 2 to 15 years. All cattle had been raised together at DWR and each herd spent at least six months together prior to the study in other pastures throughout the ranch. Both herds had grazed under light grazing pressure on extensive pastures throughout their lifetime. The study was conducted in accordance with the research protocol approved by the New Mexico State University Institutional Animal Care and Use Committee (approval number 2019-021).

### 2.3. Devices

Randomly-selected cows were fitted with a GPS tracking collars with IgotU GT-120 and IgotU GT-600 receivers [25]. Thirteen IgotU-600 receivers recorded positions at 2-min intervals. Nineteen IgotU GT-600 receivers and thirty-five IgotU GT-120 recorded positions at 10-min intervals. The difference in recording times (2 versus 10-min intervals) between the IgotU GT-120 and GT-600 receivers was based on the capacities of the attached batteries. The low stock density pasture had a total of 35 fitted GPS collars including six recording at 2-min intervals (IgotU GT-600) and 29 recording at 10-min intervals (19 IgotU 120, and 10 IgotU 600). The high stock density pasture had a total of 32 fitted GPS collars distributed with 7 recording at 2-min intervals and 25 recording at 10-min intervals (16 IgotU GT-120, and 9 IgotU GT-600). Tracking collars were attached to cows in the high stock density pasture (HSD) on 1 June 2019 and on the low stock density pasture (LSD) on 4 June 2019. Devices were removed on 31 October 2019.

### 2.4. Experimental Design

During a six-week period from 5 June to 16 July, animals were tracked to determine animal distribution and spatial movement patterns at two stocking densities. The stocking density during the trial were 0.123 animals ha^−1^ and 0.417 animals ha^−1^ for LSD pasture and HSD pasture, respectively. On 17 July, the ranch manager moved the cows from the HSD pasture to a new paddock based on a planned pasture rotation (Table 1). The herd in the LSD pasture remained in the paddock until August 12. The stocking densities were set by the ranch manager and historical grazing management.

Utilization of palatable perennial grasses (primarily Bouteloua spp.) was measured at 10 randomly placed transects in each pasture on 1–3 June, 27–28 June, and 17–20 July using height-weight relationships [26]. At each of the randomly selected transect locations in each pasture, the heights of 15 grass plants were measured at 2 m intervals. Forage mass was measured at the same locations by clipping ten, 0.25 m^2^ frames. Clipped vegetation was separated into perennial grasses, forbs, and standing dead vegetation (previous years’ growth).

### 2.5. GPS Data Processing

Raw data from the GPS units were downloaded into @Trip PC software (Mobile Action Technology, New Taipei City 23143, Taiwan). Positions that were located outside of the study pasture (e.g., GPS fixes with poor accuracy) were deleted from the data set. In addition, positions with unusual course deviations (>100°) and velocity rates that are greater than an average walking speed for cattle (84 m/min) [27] were likely inaccurate and were also removed from the data [28]. These processing steps were conducted in R (R Development Core Team, 2011) using the package ‘animaltracker’ [29] which was developed from Knight et al. [28]. If collar failure occurred before the end of the tracking period (16 July), the animal was removed from the study. A total of six and eight GPS tracking collars were removed due to collar failure from LSD and HSD, respectively. Latitude and longitude from the GPS collars were converted to Universal Transverse Mercator (UTM) coordinates using University of Wisconsin batch converter [27]. Cattle locations within 200 m of water were removed from the analyses. Associations within 200 m of water are likely resting or drinking and not grazing [17]. We were more interested in evaluating associations that occurred away from water during traveling and grazing bouts. Therefore, associations among dyads were evaluated with only GPS locations at distances farther than 200 m from water to exclude social interactions that occurred at water.

### 2.6. Analysis of Association Patterns

GPS tracking data were analyzed using the social association program ASSOC1 4.0 [30]. Utilizing UTM locations and time signatures on radiotelemetry or GPS-tracked wildlife and livestock, the ASSOC1 program evaluates associations among individual pairs, or dyads. ASSOC1 has been used to assess spatio-temporal independence of GPS- or radio-telemetry tracked bighorn sheep [31], elk [30,32,33], mule deer [33,34,35], feral goats [36], cattle [16,17,21,37], and cattle-wolf interactions [38].

Previous studies [17,21,39] have used spatial criteria to determine if cattle were moving independently or if there was a spatial association among individuals. For example, animals may be moving independently if more than 75% of the GPS locations between individuals were greater than 75 m apart. The 75 m spatial criterion indicates that cattle are within the same 1.7 ha area which in many cases could be representative of the grazing patch spatial scale [40]. Stephenson and Bailey [17] also used a 500 m spatial criteria to identify associations between individual cattle which is equivalent to a 78.5 ha area based on a 500 m radius. If two individuals were within 500 m the cattle likely would be within the same general feeding site [40] but may or may not have visual contact due to topography or vegetation structure [17]. This 500 m spatial criterion may be arbitrary, but Lendrum et al. [34] used it to assess mule deer independence. If greater than 50% of GPS locations were less than 500 m apart these authors considered the mule deer to associated and not independent. Utilizing the above justifications, the authors used the spatial criteria of 75 m and 500 m to determine independence among the animals.

The ASSOC1 program created association matrices, and the strength of the associations was calculated for each dyad using a half-weight index association measure. One of the most used measures of association is the *HWI* in evaluation association patterns between individual animals. The *HWI* has been used to understand factors that affect associations of cattle [17,41] and bison [42]. The equation to calculate *HWI* is:(1)$$HWI=\frac{X}{X+0.5\left(Ya+Yb\right)}$$
where *X* is the number of associations within specified distances (75 m or 500 m), *Ya* is the total number of points for the first individual animal, and *Yb* is the total number of points for the second individual within the dyad. A *HWI* of 1.0 would represent 100% dyadic interaction within the specified distances while a *HWI* of 0.0 would represent 0% interaction within the specified distance.

Analyses were conducted using cow locations recorded at 30-min intervals corresponding to the top (:00 min) and bottom (:30 min) of the hour. Analyzing two positions per hour was a compromise to reduce the dataset to a manageable size and allow for the evaluation of social interaction at a spatial and temporal scale similar to patch and feeding site levels (Bailey et al., 1996). For the IgotU GT-600 units recording at 2 min intervals, one position within 5-min windows at the top (:58–:02) and bottom (:28–:32) of the hour. The position closest to :00 and :30 was selected. For the IgotU GT-600 and IgotU GT-120 units (recording at 10-min intervals) positions were selected from 10-min intervals at the top (:56–:05) and bottom (:26–:35) of the hour. The position closest to :00 and :30 was selected. This process reduced the size of the dataset and the intervals allowed for at least one location to be recorded within the interval. Two-digit hour (00–23) and top (1) or bottom digit (2) was added to the Julian date to indicate a unique time stamp. For example, the time stamp for 10 July at 0200 would be 191021 (Julian date = 191, hour code = 02, top of hour = 1) for the ASSOC1 program.

Herds in the two pastures (LSD and HSD) were analyzed separately in the ASSOC1 program. Associations were calculated for the entire period between 5 June and 16 July to obtain the most and least associated dyads in each pasture. In addition, associations among tracked cows were evaluated by week. To visually interpret the variability of individuals, the mean daily distance between individuals of the most and least associated dyads in each study pasture were calculated. Twice per hour, the distance between two tracked cows (dyads) was calculated using the Pythagorean Theorem and then averaged for each day.

### 2.7. Statistical Analysis

The weekly average HWI of dyads at 75 m and 500 m were analyzed separately using the repeated measures procedure of PROC MIXED in SAS [43]. The fixed effects in the model included linear, quadratic, and cubic function of week (1 to 6), stocking density (low or high) and the interaction of week by stocking density. If the quadratic and cubic functions of week were not significant, they were dropped from the model. The subject of the repeated measures analyses was dyad (1 to 406) within stocking density. Covariance of repeated records was modelled using the autoregressive order of 1 (AR1) covariance structure. The AR1 structure was used to because the fixed effect of week was considered a continuous variable [43]. In this case study, we use dyads (cow pairs) as the experimental unit (subject of repeated measures analysis), which may potentially violate independence assumptions. However, the response variable, HWI, has been used to evaluate independence among animals and the analysis can indicate a level of independence among cows in this study.

The weekly average distances from water for each animal were analyzed using the repeated measures procedure of PROC MIXED in SAS [43]. The fixed effects in the model include linear, quadratic, and cubic function of week (1 to 6), stocking density (low or high) and the interaction of week by stocking density. If the quadratic and cubic functions of week were not significant, they were dropped from the model. The subject was animal within stocking density. Covariance of repeated records was modelled using the autoregressive order of 1 (AR1) covariance structure. Similar to the analyses of HWI, the AR1 structure was used because the fixed effect of week was considered a continuous variable [43].

## 3. Results

No differences between stocking densities were detected in association strength between cows at the 75 m distance criterion (*p* = 0.22). Across the six-week study, the mean HWI for the 75 m distance criterion in the high stocking density (HSD) pasture and low stocking density (LSD) pasture were 0.023 ± 0.001 SE and 0.025 ± 0.001 SE, respectively (Figure 1). There was a linear (*p* = 0.016) effect of week (Figure 1). Association strength declined over the study (−0.0006 ± 0.0003 SE HWI units per week). No interaction of week by stocking density was detected (*p* = 0.87).

For association strength at 500 m, there was an interaction (*p* < 0.001) of stocking density and week. Throughout the study, weekly HWI decreased more rapidly within the HSD pasture compared to the LSD (Figure 2). Across the six-week study, the mean HWI at 500 m was 0.1824 and 0.1627 for the HSD pasture and LSD pasture, respectively. Association strength declined over the study. For the linear component of the relationship was −0.0144 ± 0.003 SE HWI units per week for the low stocking density and −0.0160 ± 0.003 SE HWI units per week for the high stocking density. The rate of decrease was less at the end of the study (Figure 2), a quadratic response (0.0016 ± 0004 SE HWI units per week squared).

The HWI of all dyad associations at the 75 m distance criterion were less than 0.10. The highest HWI was 0.08 for the LSD pasture and 0.07 for the HSD pasture. All of the dyad associations at the 500 m distance were all less than 0.30. The highest HWI were 0.249 and 0.288 for the LSD pasture and HSD pasture, respectively. The average daily mean distances between the most associated cows in the LSD pasture was 789 m with mean distances on 13 of the 42 days of the study being less than 500 m. The average daily mean distance between the most associated cows in the HSD pasture was 1076 m with 7 of the 42 days being less than 500 m. The distance between the most associated dyad at 500 m (Figure 3) in the low stocking density pasture increased over the duration of the study while the distance between the most associated in high stocking density pasture decreased. The distance between the least associated dyad using the 500 m criterion (Figure 4) in both pastures increased faster in the HSD pasture faster than the LSD pasture.

There was a cubic response (*p* < 0.001) with linear and quadratic differences for distance from water over the six-week trial. Distance cattle travelled from water increased at the end of the trial (Figure 5). Distance to water was greater (Figure 5) (*p* < 0.0001) in the larger pasture (low stocking density) and an interaction of stocking density by week was detected (*p* < 0.001). As expected, cattle travelled farther from water in the larger pasture (Figure 5). Across the six-week study, the mean distance from water were 861 m and 2295 m for the HSD and LSD pasture, respectively.

Forage utilization in the HSD and LSD pastures was 5% and 10%, respectively, and the forage mass was 2601 kg ha^−1^ and 2343 kg ha^−1^ at the beginning of the study, respectively (Figure 6). At the end of the study, forage utilization was 24% and 20% and forage mass was 1828 kg ha^−1^ and 1904 kg ha^−1^ in the HSD and LSD pasture, respectively.

## 4. Discussion

Appropriate stocking levels and achievement of desired livestock distribution (often uniform distribution) are critical to rangeland management to ensure that forages are not grazed excessively. It is time consuming for managers to inspect rangeland to determine when to move livestock to a new pasture [3]. In this case study, spatial associations among cows change as pastures become defoliated, which suggests this metric has potential to help monitor pasture conditions. Livestock production in extensive systems typically allows livestock to express their natural behavior [3]. By monitoring normal behaviors, caretakers have the potential of adjust grazing management plans based on the distance livestock travel from water or changes in time spent grazing, traveling, or resting.

Livestock distribution and potentially cattle social associations in the western US can depend on abiotic factors such as topography, water availability, and thermal cover, and biotic factors, such as forage quantity and quality [9]. Many of these factors affect livestock differently across regions. In a hot semi-arid region, studies have found that during the warm growing season cattle herd dispersion is greater than during the cold season [21,22,44,45]. However, in northern, mountainous regions and other arid regions, cattle tend to travel and forage in more compact groups during the growing season and disperse during the winter when resources are sparse [20,23,46,47,48]. Rough topography and landscape obstacles may influence associations among subgroups. Stephenson and Bailey [17] found that cows on southern short grass prairie in central New Mexico, a site with terrain similar to DWR, had no HWI above 0.25 and 0.50 at 75 m and 500 m criteria, respectively. Clark et al. [49] considered cattle behaving independently when they spent more than 75% of their time at distances greater than 75 m. Cattle in our study at the DWR appeared to distribute themselves independently based on this 75% of the time at greater than 75 m apart criterion. The low HWI of even the most associated dyads at 75 m suggests that cattle may have chosen grazing patch sites without strong influences from other GPS tracked cattle in the study. Often, cattle will graze distinct separated areas due to the extensiveness of some rangelands [47,48,50,51]. Stephenson and Bailey [17] reported that some cattle utilized distinctly different areas of an extensive 9740 ha pasture in southern Arizona and never were within 2500 m of each other over a three-month tracking period. Cows in this study developed into different subgroups of cattle separated by a large ridge even though all cattle were turned out at the same time and location. The rugged terrain likely increased associations of cattle within subgroups using a particular portion of the mountainous pasture while segregating the subgroup with others in the herd.

Pasture size and herd size can affect the independence of animals. Stephenson and Bailey [17] found completely independent cows (i.e., HWI = 0.0 at 500 m) with a herd size of 250 cows grazing 9740 ha in southern Arizona. In contrast, Bailey et al. [52] visually observed that cows were never more than 100 m apart in a herd size of only 10 cows on 3 large pastures (>1000 ha) in the Chihuahuan Desert of southern New Mexico. Cheleuitte-Nieves et al. [21] selected 10 individuals from a herd of 31 animals on a study site of 100 ha where animals spent 70% of their time within 200 m of each other. Harris et al. [20] also observed that small herds (herd size < 20) tended to associate as a group with greater than 50% association at 100 m spatial distances. Stephenson et al. [16] concluded that the associations of cows in herds less than 40 cows may not reflect the patterns observed in larger herds (50 to 200 cows). Cows in this study appeared to behave independently as no dyad HWI were above 0.25 or 0.50 for the 75 m and 500 m, respectively, the criteria described by Stephenson and Bailey [17]. Similar to Stephenson et al. [16] and Stephenson and Bailey [17], GPS tracked cows in this study did not appear to form strong associations and did not typically show fidelity to a group. Even the most associated cows in both stocking densities were on average at least 500 m apart for over 2/3 of the study. This could be the result of random selection of cattle to be tracked within the herds and not capturing small groups of highly associated cows [17] or the fusion-fission dynamics of a herd where cattle come together and separate with different herd mates at different times during grazing [48]. However, the large sample of cows tracked in both the LSD (35 out of 135 cows) and HSD (32 out of 130 cows) pastures makes this less likely.

Animal interactions can affect subgrouping [14]. During the study, animal behavior was visually observed by the research team. When cattle went to drink at mid-day, many cattle would enter the watering lot and interact with other cattle that were drinking. During many interactions, two or more cattle and calves would hook each with their horns while at the drinker. This aggressive behavior could affect the subgrouping of animals and modify social associations. The aggressive nature of the cattle may have decreased social associations through these interactions. Changes in social associations may encourage individual cows to graze with other animals and increase the distance from other cows during foraging.

Livestock can graze at sustainable levels using proper stocking rates, but heterogeneous landscapes (e.g., variable topography and plant communities) often result in uneven grazing pressure with areas of high and low foraging in the same pasture [9]. During the grazing period, forages in preferred areas may be depleted requiring animals to disperse to seek out alternative forage. We observed a decreased mean HWI for cattle in both the LSD and HSD pastures over the six weeks of the study suggesting that collared animals within the herd were farther than 75 m from each other for longer amounts of time. This could indicate that animals are dispersing farther to find suitable grazing areas as forage becomes limited in preferred areas. A sustained decrease in HWI could indicate dispersion of animals potentially due increased levels of defoliation and decreased forage resources. Sato [53] concluded that vegetation condition directly influenced the dispersive movement of grazing. At the 75 m spatial distance criterion, associations in both stocking densities decreased linearly at similar rates during the study. As cattle searched for preferred forage, cows in both pastures may have moved further apart and dispersed farther during the latter part of the study when forage availability was less abundant. Manning et al. [54] observed shifts in cattle movement patterns and grazing behavior in response to changes in forage availability monitored using satellite imagery. Associations at the 75 m (or similar distances) may have value for determining the extent that cattle search for forage, which could help managers determine when to end grazing in a pasture.

Additional research is needed to verify the results observed in this case study. The difference in pasture size in this study was a limitation, because the maximum distance to water was over two time greater in the larger pasture, which would affect grazing distribution patterns. Associations at the 500 m criterion in this study were affected by pasture size and may not be useful for monitoring cattle dispersion except in large pastures. In pastures that are less than 500 ha, there are fewer areas where cattle can separate by more than 500 m (compared to pastures greater than 500 ha). Additionally, paddock size can influence distance travelled by cattle with greater distance travelled per day in large paddocks compared to smaller paddocks [55]. However, the weekly changes in associations at the 75 m criterion and changes in average distance travelled from water were relatively similar in both pastures, which suggests that pasture size has less effect on cattle dispersion at a smaller spatial scale (e.g., 75 m association criterion). The weekly changes in the distances between the most and least associated dyads based on the 500 m criterion may also be informative for pasture management. The distance between the most associated animals decreased in the HSD pasture and increased in the LSD pasture. This decrease in distances of animals grazing under higher stocking densities could be due to the reduction of forage availability requiring cows to travel further from water which would result in more time in same part of the smaller pasture. The higher availability of forage in the LSD pastures may have allowed the most associated dyad to spend more time together over time as the two cows potentially became more familiar as the study progressed. The distance between the least associated dyads both increased during the study. This increase in distance between the least associated dyads could indicate separation of the subgroups in an effort to locate forage.

## 5. Conclusions

Forage utilization levels typically increase more rapidly at higher stocking densities than at lower stocking densities, which forces managers to make more timely decisions to move between pastures at higher stocking densities. Social interactions as monitored by spatial positions among individual cows may provide some insight into cattle grazing patterns, forage utilization and availability. Cattle dispersed as forage utilization increased and forage mass decreased over time. Improvement of technologies and associated software could give producers near-real time cattle tracking data, which could be used to monitor changes in social association and help assess defoliation levels of vegetation. Such remote monitoring approaches would assist rancher to make more timely grazing management decisions and help improve rangeland health and sustainability. However, more research is needed to verify the changes in social interactions that occur as forage is depleted and perhaps use spatial relationships among cows to help remotely monitor forage utilization and availability. In this case study, pasture size and stocking density were confounded, but smaller pasture size is commonly associated higher stocking densities. Additional research is needed to better understand the impacts of stocking density and pasture size on cattle social interactions and associations.

## Figures and Tables

**Figure 1 animals-11-02635-f001:**
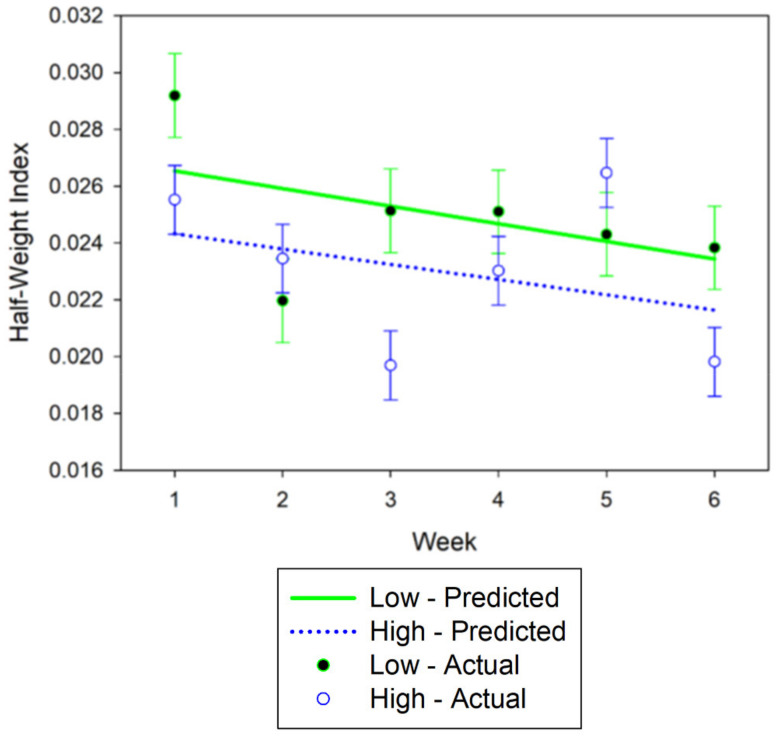
Weekly half-weight indices (HWI) means at 75 m spatial criterion for cows in the high stocking density (North Ditch) pasture and low stocking density (North) pastures. The predicted regression equation is defined by the lines and weekly means are shown as points. Error bars are standard errors.

**Figure 2 animals-11-02635-f002:**
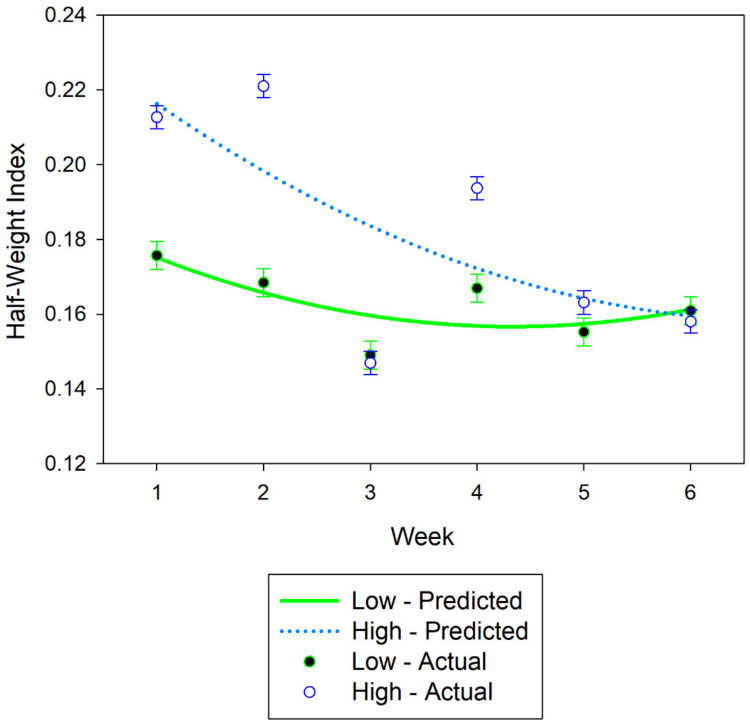
Weekly half-weight indices (HWI) means at the 500 m spatial criterion for cows in the high stocking density (North Ditch) and low stocking density (North) pastures. The predicted regression equation is defined by the lines and weekly means are shown by points. Error bars represent standard errors.

**Figure 3 animals-11-02635-f003:**
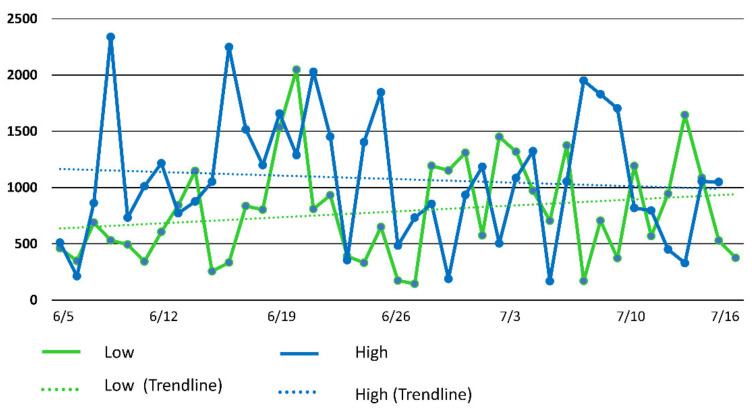
Average distance between the most associated dyads each day (using 500 m spatial distance criterion) from both low stocking density (North) and high stocking density (North Ditch) pastures. The trend line displays the linear regression of distance on day.

**Figure 4 animals-11-02635-f004:**
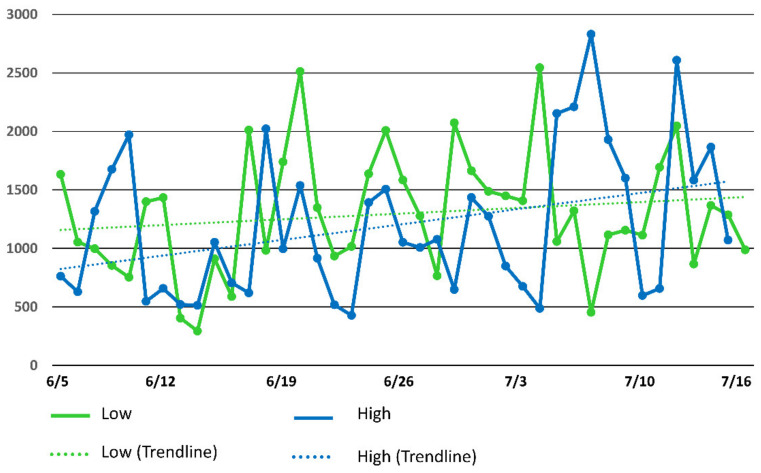
Average daily distance between the least associated dyads (using the 500 m spatial distance criterion) in the low stocking density (North) and high stocking density (North Ditch) pastures. The trend line displays the linear regression of distance on day.

**Figure 5 animals-11-02635-f005:**
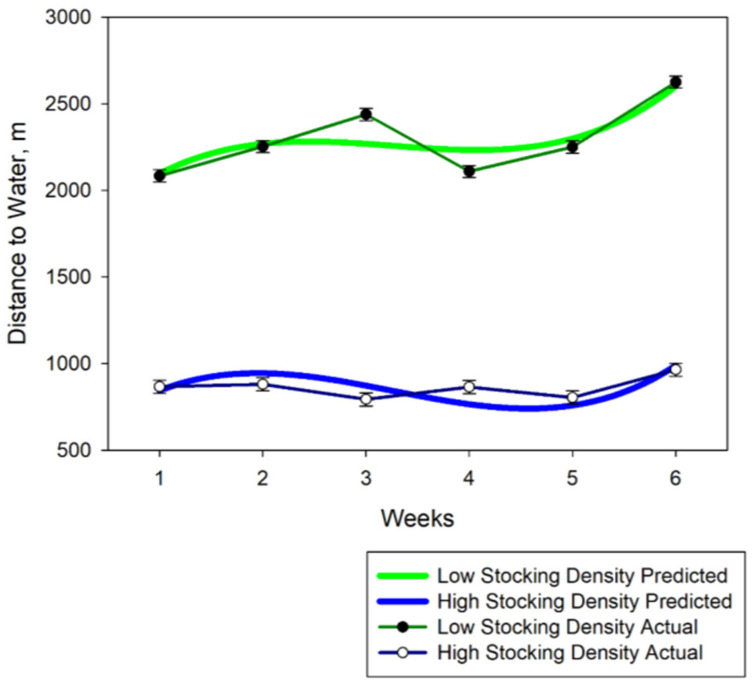
Average distance from water by week for both the high stocking density (North Ditch) and low stocking density (North) pastures. Predicted values from regression equation are solid lines and the mean distances each week are points. Error bars represent standard errors.

**Figure 6 animals-11-02635-f006:**
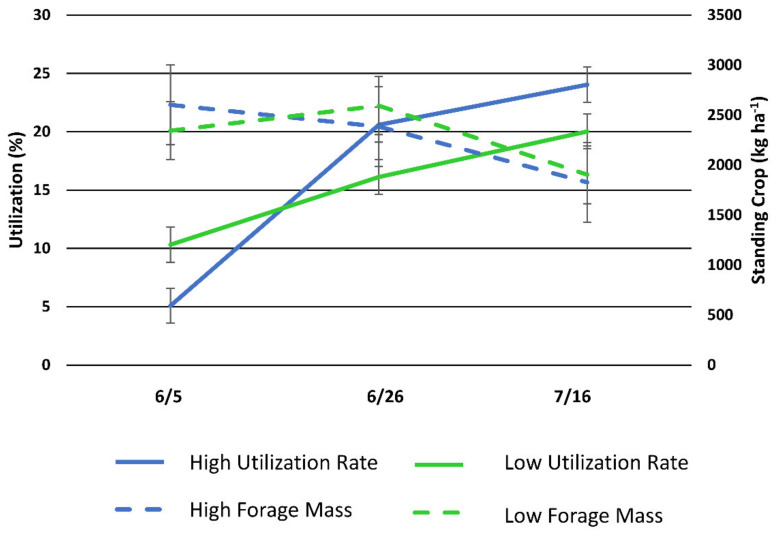
Mean forage utilization and herbaceous forage mass plus and minus standard error (error bars) of the (high stocking density (North Ditch) and low stocking density (North) pastures during the 6-week study.

**Table 1 animals-11-02635-t001:** Periods of cattle tracking, study dates, pasture descriptions, number of GPS tracked animals at the Deep Well Ranch (DWR).

Pasture	Start Date–End Date	Length of Tracking (d)	Pasture Size (ha)	Water Source	Herd Size (Pairs)	Maximum Distance from Water (m)	Number of GPS-Tracked Cows (*n*)	Number of Dyadic Associations ^1^
High stock density (North Ditch)	5 June–16 July	42	312	1	130	2100	24	276
Low stock density (North)	5 June–16 July	42	1096	1	135	4500	29	406

^1^ number of dyadic associations n × (n − 1)/2.

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
