# Peer review of "Temporal Changes in Association Patterns of Cattle Grazing at Two Stocking Densities in a Central Arizona Rangeland"

_animals, 2021, doi:10.3390/ani11092635_

Round 1

Reviewer 1 Report

I recommend this manuscript for publication, following corrections and revision points (attached) have been addressed. 

Author Response

Review for Animals: Tobin et al. Identify changes in livestock social associations and spatial

location at 2 stocking densities over a 6-week grazing period: temporal changes in cattle

associations; detection using GPS tracking.

Line 19-20 – ? forage mass and utilisation both increased? Subsequent reading of the main

manuscript reveals forage mass biomass decreased while utilisation increased.

            Revised to “Forage mass decreased throughout the study as forage utilization increased from 5 to 24%”

Line 24-25 – rangeland health and livestock health degradation avoidance. The authors did

not seem to specifically discuss this or develop this argument much in the discussion or

conclusions section. Could the authors expand on this statement in the summary, by

elaborating on health of rangeland and livestock, from the implications of their research

findings in this case study?

Added a section in the conclusion explaining how monitoring social association could be used to make more timely grazing management decisions and improve rangeland health.  We also removed the reference to livestock health since it is not addressed in this study.

Line 54 – large gap in text between ‘to’ and ‘intensive’. Amend.

            Revised as requested

Line 55-57 – Are references 3-6 the only literature linking human influence and

management of rangelands on grazing animal behaviour for improved watershed health? So

far, the authors have referred to their own work a lot.

These papers more specifically support these statements.  Other papers are less focused on these issues and findings.

Line 58 – An ambiguous start. Livestock grazing distribution ? Even, bunched, controlled?

Needs to be qualified in the context of grazing management principles.

Vallentine (2000) does not place a qualifier for grazing distribution since the type of distribution management (bunched, dispersed, etc.) depends on the situation.  We modified this statement to clarify.

Vallentine, J. F. 2000. Grazing Management. San Diego, CA: Academic Press.

Line 461:Reference 11 – source? Book, journal?

            Revised as requested – Proceedings Paper

Check all references have complete descriptions. E.g. 12, Line 462, 463.

Line 71 – insert a comma after [17].

            Revised as requested

Line 480 – should be Rang. Ecol. Management? Check.

            Correct – Revised as requested

Line 82 – insert ‘of’ between ‘indicators’ and ‘livestock’.

            Revised as requested

Line 142 – space needed between June and 1-3.

Revised as requested

Line 145 – Amend to 0.25m2. Check for correct formatting throughout the manuscript.

            Revised as requested

Line 160 – check grammar.

Line 524 – correct title? Mechanisms that result in….

            Correct, Revised as requested

Line 170-181 – This section seems redundant. What spatial criteria did these authors use for

this study? If any, 75m or 500m? Both. Need a statement clearly saying what spatial criteria

this study used. Reading further, it seems both are used. But this was not made clear,

explicit in the methods.

            Justification was place on line 182-183

Figures 1 and 2: why use the terms Heavy and Light, instead of High Density and Low

Density, as per the rest of the manuscript? Consistency needed here. High SD, Low SD.

Check.

Revised as suggested,

Line 310 – optimal, even distribution of livestock; qualify what livestock distribution means

in this context.

Revised as suggested.

Line 311 – grammar check.

Addressed.

Line 316 – in text, decrease spacing. Check all of the manuscript for gaps like this.

            Revised as requested

Line 317 – grammar check.

            Revised as requested

Line 322-326 – Is the magnitude of these dispersal differences greater or less, the

same/similar between hot semi-arid and mountainous and other arid regions? Is the

magnitude of dispersal the same between climatic regions?

            The dispersal differences would be due to several factors including, size of pasture, topography limitations, etc. It would be difficult to quantify this without the in depth information of each location from the studies.

Line 367 – grammar check.

Revised

Line 371 – increase distance for, or from other cows.

            From – Amened as requested

Line 364-371 – Any supporting references here ? This paragraph currently reads as

speculation only.

            The observations were anecdotal but supporting references were added as requested

Line 380 – very possible occurrence. Are there no supporting references of this point, from

foraging ecology literature?

            Supportive references added “Sato [53] concluded that vegetation condition directly influenced the dispersive movement of grazing”

Line 382 – grammar check.

Revised

Line 401-403 – suggest inserting ‘foraging’ before ‘in the same part…’

Revised as suggested

Line 420 – were confounding variables, or were confounded by each other? Suggest

elaboration to improve clarity of this point.

Revised

Lines 421, 422 – grammar check.

Revised

Reviewer 2 Report

I have reviewed the manuscript titled “Temporal changes in association patterns of cattle in two stocking density scenarios in a central Arizona rangeland” submitted to Animals. The manuscript describes a case study analyzing the association patterns among cow-calf pairs under two stocking rates and their relationship with pasture utilization and changes in forage mass. The manuscript represents a valuable piece of work for rangeland management and falls within the scope of the journal. There are some aspects of terminology, methodology and structure that would need to be improved.

General comments

According to Allen et al. (2011) “An international terminology for grazing lands and grazing animals” (doi: 10.1111/j.1365-2494.2010.00780.x), the correct term to be used for this study is “stocking rate” and not “stocking density”:

  • Stocking rate: the relationship between the number of animals and the total area of the land in one or more units utilized over a specified time; an animal-to-land relationship over time.
  • Stocking density: the relationship between the number of animals and the specific unit of land being grazed at any one time; an instantaneous measurement of the animal-to-land area relationship.

Therefore, I strongly recommend you replace the terms throughout the document. This would also affect the acronym for groups, i.e., HSR and LSR.

Similarly, the preferred term for the amount of forage harvested in a given area is “forage mass” and not “standing crop”:

  • Forage mass: the total dry weight of forage per unit area of land above a defined reference level, usually ground level, at a specific time. The term forage mass is preferred to alternatives like ‘standing crop,’ ‘forage yield’ and ‘available forage,’ which involve assumptions (often unspecified) about canopy characteristics and harvesting procedures.

In this study, the contrasting stocking rates were achieved by modifying the pasture size, which has an effect on livestock distribution per se. Another option would have been to modify the herd size and use similar pasture sizes. This limitation in the design has not been discussed properly. I suggest including the discussion of this confounder factor at the beginning of the Discussion section.

Specific comments

 Title

  • Please use “stoking rate” as explained above.
  • The term “scenario” is usually used for predicted or hypothetical conditions (e.g. future climate scenarios, different scenarios in a sensitivity analysis). I’d suggest the title “Temporal changes in association patterns of cattle at two stocking rates in a central Arizona rangeland”

Simple summary

  • L19-21: In L19-20 two variables are mentioned (standing crop and forage utilization), however, the range of values presented corresponds only to the change in time of the latter. Please amend.

Introduction

  • There is a recent review (doi: /10.3390/s21082696) that has analyzed the factors affecting cattle distribution assessed with GPS tracking. It may be interesting to reinforce some ideas. Just a suggestion.
  • L86: I’d suggest not using “scenarios” in this manuscript. If you accept my suggestion, please remove the term throughout the manuscript.
  • L85-95: I believe the Introduction section should end with the objectives, and in some case with a brief mention to the methodological approach. In this manuscript, you have added additional background after the objectives. I believe this should have been presented earlier in the document. Please restructure accordingly.

Material & Methods

  • The number of available GPS units seem not to match the number of units used: 32 IgoU GT-600 (13 at 2 min and 19 at 10 min), plus 25 IgotU GT-120. So, in total you had 57 units. However, you used 67 (35 in the Low and 32 in the High). Please check these figures.
  • L115-116: Should the readers assume that the animals spent 6 months in different fields than the ones used in the experimental period? This should be explicitly stated. Moreover, the size of those fields should be included since the size or characteristics of those fields could have affected their behavior in the experimental fields.
  • L121: Is it “IgotU GT-600” instead?
  • L135: How were the stocking rate values to be tested defined? From previous experiments? From usual management of those two pastures?
  • Table 1: please make the table wider to adjust text in fewer lines. The heading for the last column has a uppercase “1” but there is no foot note below the table. Please add it.
  • L142: How was the number of transect used defined (10)? Was that number enough to assess a 300 or 1100 ha pasture? Given the noticeable difference pasture size (the “Low” is more than three times bigger than the “High”), why the authors sed the same number of transects? Wouldn’t have been better to increase the number of transects proportionally to the size to better capture the spatial variability of forage mass and forage utilization?
  • L157: It seems that 8 GPS tracking collars were removed for the “High” from the study (that is the difference between 24 in Table 1 and 32 in L127 for the “High”). Please check.
  • L160: Even if the animals are not grazing near the water points (which would be arguable), they could still be associated if they are within 75m distance from each other. The association in this manuscript has not been defined in terms of a specific activity (i.e. grazing). Please support this exclusion with arguments and references. Otherwise, the analysis should be redone to include the fixes near the water points.
  • A map showing the layout f the two pastures used with the features (e.g. water points) would be useful to understand the experimental conditions.
  • L167-181: all this justification of the ASSOC1 methodology is not part of the M&M section. If relevant or the interpretation of results, this should be moved to the Discussion section. Otherwise, it should be removed.
  • L183-184: “The ASSOC1 program created association matrices and the strength of the associations was calculated for each dyad using a Half-Weight Index association measure”
  • L184-187: “One of…..[42]” this is not part of the M&M section. If relevant or the interpretation of results, this should be moved to the Discussion section. Otherwise, it should be removed.

Results

  • Figure 1 to 6 should be consistent in their legend. The authors interchangeable use “light”, “heavy”, “low”, “high”. I suggest using “Low stocking rate” and “High stocking rate” for all the tables. You could use one “Low” and “High” provided you define these two legends in the footnote of the figure.
  • L281-283 and L287-289: all these descriptions of results should be removed. It should be presented only in the main text (already done)

Discussion

  • Please start this section discussingthe confounding factor “pasture size” (please see General comments).
  • L364-365: the observations of behavior are not described in the M&M section. Please include the methodology for these observations in the relevant section. Otherwise, remove this element from the Discussion (it seemed to be anecdotal only).

Conclusion

  • This last section must only be supported by the results. Therefore, finding from other studies must not be presented (e.g. L410-412, 414-419). Please rephrase this section completely to present appropriate conclusions based on your results.

Author Response

Comments and Suggestions for Authors

I have reviewed the manuscript titled “Temporal changes in association patterns of cattle in two stocking density scenarios in a central Arizona rangeland” submitted to Animals. The manuscript describes a case study analyzing the association patterns among cow-calf pairs under two stocking rates and their relationship with pasture utilization and changes in forage mass. The manuscript represents a valuable piece of work for rangeland management and falls within the scope of the journal. There are some aspects of terminology, methodology and structure that would need to be improved.

General comments

According to Allen et al. (2011) “An international terminology for grazing lands and grazing animals” (doi: 10.1111/j.1365-2494.2010.00780.x), the correct term to be used for this study is “stocking rate” and not “stocking density”:

  • Stocking rate: the relationship between the number of animals and the total area of the land in one or more units utilized over a specified time; an animal-to-land relationship over time.
  • Stocking density: the relationship between the number of animals and the specific unit of land being grazed at any one time; an instantaneous measurement of the animal-to-land area relationship.

Therefore, I strongly recommend you replace the terms throughout the document. This would also affect the acronym for groups, i.e., HSR and LSR.

            The authors value this suggestion, but we prefer stocking density. The purpose of the two stocking densities is to achieve two different rates of defoliation of forages. When considering the study, the grazing within the north ditch pasture ends with the study due to the animals being moved onto a new pasture while grazing in the low stocked pasture continued. The ranch manager’s grazing management goal is to graze all pastures to a light (25-30% utilization rate) stocking rate. Using the term stocking rate is not precise because the grazing period in each pasture was not equal.  The overall stocking rate was similar in both pastures because the grazing periods were not equal.  Stocking density is a more precise term that describes that the defoliation rate differs between the two pastures.

Similarly, the preferred term for the amount of forage harvested in a given area is “forage mass” and not “forage mass”:

  • Forage mass: the total dry weight of forage per unit area of land above a defined reference level, usually ground level, at a specific time. The term forage mass is preferred to alternatives like ‘forage mass,’ ‘forage yield’ and ‘available forage,’ which involve assumptions (often unspecified) about canopy characteristics and harvesting procedures.

Revised as requested

In this study, the contrasting stocking rates were achieved by modifying the pasture size, which has an effect on livestock distribution per se. Another option would have been to modify the herd size and use similar pasture sizes. This limitation in the design has not been discussed properly. I suggest including the discussion of this confounder factor at the beginning of the Discussion section. 

We agree that we could have used the same pasture size and different numbers of animals.  However, previous research (Stephenson et al., 2016) demonstrated the impact of herd size on associations among animals.  Our primary interest was the change in associations among cows in response differing forage utilization levels so having different pasture sizes and similar herd sizes was a better approach to help answer our questions. We added language in the Discussion to address the limitation of the study. See lines 406 to 417.

Specific comments

 Title

  • Please use “stoking rate” as explained above. 
  • The term “scenario” is usually used for predicted or hypothetical conditions (e.g. future climate scenarios, different scenarios in a sensitivity analysis). I’d suggest the title “Temporal changes in association patterns of cattle at two stocking rates in a central Arizona rangeland”

Title was revised and “scenario” was deleted.  However, we contend that stocking density is a better term than stocking rate for this study.  See above.

Simple summary

  • L19-21: In L19-20 two variables are mentioned (forage mass and forage utilization), however, the range of values presented corresponds only to the change in time of the latter. Please amend.

Revised as requested

Introduction

  • There is a recent review (doi: /10.3390/s21082696) that has analyzed the factors affecting cattle distribution assessed with GPS tracking. It may be interesting to reinforce some ideas. Just a suggestion.

Although this recent review is interesting, it really doesn’t add anything new regarding the impact of social interactions on grazing distribution that are not already discussed in the paper.

  • L86: I’d suggest not using “scenarios” in this manuscript. If you accept my suggestion, please remove the term throughout the manuscript.  

Revised as suggested.

  • L85-95: I believe the Introduction section should end with the objectives, and in some case with a brief mention to the methodological approach. In this manuscript, you have added additional background after the objectives. I believe this should have been presented earlier in the document. Please restructure accordingly.

Revised as suggested

Material & Methods

  • The number of available GPS units seem not to match the number of units used: 32 IgoU GT-600 (13 at 2 min and 19 at 10 min), plus 25 IgotU GT-120. So, in total you had 57 units. However, you used 67 (35 in the Low and 32 in the High). Please check these figures.

67 units is the correct amount.  It was 35 IgotU GT-120 units rather than 25.

  • L115-116: Should the readers assume that the animals spent 6 months in different fields than the ones used in the experimental period? This should be explicitly stated. Moreover, the size of those fields should be included since the size or characteristics of those fields could have affected their behavior in the experimental fields

Recommendations were input into the manuscript “. All cattle had been raised together at DWR and each herd spent at least six months together prior to the study in other pastures throughout the ranch. Both herds had grazed under light grazing pressure on extensive pastures throughout their lifetime.”.

  • L121: Is it “IgotU GT-600” instead?

Revised as requested

  • L135: How were the stocking rate values to be tested defined? From previous experiments? From usual management of those two pastures?

The stocking densities are set by the ranch manager who separates his cows into 2 herds, which is the normal managemen..  The ranch manager runs his herds separately due to the AZ HWY 89  and ha,  A sentence was added explaining their management.

  • Table 1: please make the table wider to adjust text in fewer lines. The heading for the last column has a uppercase “1” but there is no foot note below the table. Please add it.

Revised as requested

  • L142: How was the number of transect used defined (10)? Was that number enough to assess a 300 or 1100 ha pasture? Given the noticeable difference pasture size (the “Low” is more than three times bigger than the “High”), why the authors sed the same number of transects? Wouldn’t have been better to increase the number of transects proportionally to the size to better capture the spatial variability of forage mass and forage utilization?

The number of transects were the same in each pasture to keep the number of samples in the low and high stocking density pastures equal. With the inherent variation in forage mass, small scale spatial variation in these pastures is often larger than the larger spatial variation because the ecological sites (primarily soils and topography) in the pastures were similar. Note that the standard errors for utilization and forage mass were relatively similar in both pastures.  Our vegetation sampling was also constrained by time and we wanted to get a minimum number of transects in each pasture.

  • L157: It seems that 8 GPS tracking collars were removed for the “High” from the study (that is the difference between 24 in Table 1 and 32 in L127 for the “High”). Please check.

Correct, revised as requested

  • L160: Even if the animals are not grazing near the water points (which would be arguable), they could still be associated if they are within 75m distance from each other. The association in this manuscript has not been defined in terms of a specific activity (i.e. grazing). Please support this exclusion with arguments and references. Otherwise, the analysis should be redone to include the fixes near the water points.

Included verbiage ”We were more interested in evaluating associations that occurred away from water during traveling and grazing bouts; therefore, associations among dyads were evaluated with only GPS locations at distances farther than 200 m from water to exclude social interactions that occurred at water.” Stephenson et al. (2017) also conducts analysis similarly to exclude interactions at watering areas.  With each pasture containing only one watering area animals would water and rest during the hottest parts of the day after their first grazing period.

  • A map showing the layout f the two pastures used with the features (e.g. water points) would be useful to understand the experimental conditions.

The authors value this suggestion, but we feel this will not add value to the manuscript beyond the information provided in Table 1.  If the reviewer does not agree with this, we will revise as requested.

  • L167-181: all this justification of the ASSOC1 methodology is not part of the M&M section. If relevant or the interpretation of results, this should be moved to the Discussion section. Otherwise, it should be removed.

The ASSOC1 methodology is included in the M&M section as a method for clarity.  We believe that all of this justification is needed if other researchers would replicate our experiment.

  • L183-184: “The ASSOC1 program created association matrices and the strength of the associations was calculated for each dyad using a Half-Weight Index association measure”

Revised as requested

  • L184-187: “One of…..[42]” this is not part of the M&M section. If relevant or the interpretation of results, this should be moved to the Discussion section. Otherwise, it should be removed.

                        The ASSOC program, spatial criterion (75 and 500 m) and half-weight index are critical for the reader to understand the data and results.  By placing this information in the Methods, readers can better understand the methodology and background of this technique without having to find out about this needed information later in the paper.  Our experience is that many livestock researchers are not familiar with association types of analyses and this information helps them better understand the process and results.

Results

  • Figure 1 to 6 should be consistent in their legend. The authors interchangeable use “light”, “heavy”, “low”, “high”. I suggest using “Low stocking rate” and “High stocking rate” for all the tables. You could use one “Low” and “High” provided you define these two legends in the footnote of the figure.

     The figures were revised as requested, except that we prefer the more precise term stocking density rather than stocking rate.

  • L281-283 and L287-289: all these descriptions of results should be removed. It should be presented only in the main text (already done)

  Revised as requested.

Discussion

  • Please start this section discussing the confounding factor “pasture size” (please see General comments).
  • L364-365: the observations of behavior are not described in the M&M section. Please include the methodology for these observations in the relevant section. Otherwise, remove this element from the Discussion (it seemed to be anecdotal only). 

The observations were anecdotal, but we feel that the aggressive behavior may affect the subgrouping of the animals and must be placed within the discussion. 

Conclusion

  • This last section must only be supported by the results. Therefore, finding from other studies must not be presented (e.g. L410-412, 414-419). Please rephrase this section completely to present appropriate conclusions based on your results

                        Revised as suggested.
